# CoPL:Parameter-Efficient Collaborative Prompt Learning for Audio-Visual Tasks

Yihan Zhao
Xi'an Jiaotong University
Xi'an, China
zhaoyihan@stu.xjtu.edu.cn

Wei Xi*
Xi'an Jiaotong University
Xi'an, China
xiwei@xjtu.edu.cn

Yuhang Cui
Xi'an Jiaotong University
Xi'an, China
yuhangcui@stu.xjtu.edu.cn

Gairui Bai
Xi'an Jiaotong University
Xi'an, China
grbai2018@stu.xjtu.edu.cn

Xinhui Liu
Xi'an Jiaotong University
Xi'an, China
liuxinhui@stu.xjtu.edu.cn

Jizhong Zhao
Xi'an Jiaotong University
Xi'an, China
zjz@mail.xjtu.edu.cn

## Abstract

Parameter-Efficient Fine Tuning (PEFT) has been demonstrated to be effective and efficient for transferring foundation models to downstream tasks. Transferring pretrained uni-modal models to multi-modal downstream tasks helps alleviate substantial computational costs for retraining multi-modal models. However, existing approaches primarily focus on multi-modal fusion, while neglecting the modal-specific fine-tuning, which is also crucial for multi-modal tasks. To this end, we propose parameter-efficient **Co**llaborative **P**rompt **L**earning (**CoPL**) to fine-tune both uni-modal and multi-modal features. Specifically, the collaborative prompts consist of modal-specific prompts and modal-interaction prompts. The modal-specific prompts are tailored for fine-tuning each modality, while the modal-interaction prompts are customized to explore inter-modality association. Furthermore, prompt bank-based mutual coupling is introduced to extract instance-level features, further enhancing the model's generalization ability. Extensive experimental results demonstrate that our approach achieves comparable or higher performance on various audio-visual downstream tasks while utilizing approximately 1% extra trainable parameters.

## CCS Concepts

• **Information systems** → **Multimedia information systems**; • **Computing methodologies** → *Artificial intelligence.*

## Keywords

Audio-Visual Learning, Prompt Learning, Multi-modal Fusion

**ACM Reference Format:**
Yihan Zhao, Wei Xi, Yuhang Cui, Gairui Bai, Xinhui Liu, and Jizhong Zhao. 2024. CoPL:Parameter-Efficient Collaborative Prompt Learning for Audio-Visual Tasks. In *Proceedings of the 32nd ACM International Conference on*

*Corresponding author

*Multimedia (MM '24), October 28-November 1, 2024, Melbourne, VIC, Australia.* ACM, New York, NY, USA, 10 pages. https://doi.org/10.1145/3664647.3681492

## 1 Introduction

Large-scale pretrained models have experienced notable progress in recent years [2, 33, 34, 41]. While fully fine-tuning pretrained models for downstream tasks can achieve commendable performance, the expensive computational cost renders it impractical for most researchers. Consequently, investigators tend to explore parameter-efficient fine-tuning (PEFT) methods [15–18, 25], aiming to achieve excellent performance with limited trainable parameters, thereby reducing the computational cost. These PEFT paradigms such as Adapter [16, 38], Prompts [18, 25, 47], and LoRA [15, 17], have demonstrated remarkable efficiency and effectiveness. They transfer the foundation models to downstream tasks by attaching a few extra parameters while keeping the pretrained models frozen.

Most PEFT-based researches focus on transferring pretrained ViT [4] or CLIP [34] to few-shot or zero-shot uni-modal tasks [12, 14, 18, 19, 53, 54]. There are still lacking profound exploration in transferring pretrained uni-modal model to multi-modal task, especially in audio-visual domain. Transferring pretrained uni-modal models to multi-modal downstream tasks is promising for two reasons. First, the optimizing objective of CLIP [34] or AudioCLIP [11] is to align the multi-modal semantic features, which means that extending to another modality requires retraining from scratch. Existing freely large-scale uni-modal foundation models could alleviate substantial computational costs and huge paired data for retraining multi-modal models. Second, the contribution of each modal to different multi-modal tasks is peculiar and irreplaceable. Transferring uni-modal models to multi-modal tasks can provide both modal-specific and modal-interaction information.

Existing methods typically focus on multi-modal fusion by introduce additional adapter branches, due to the fact that pre-trained uni-modal models lack mutual interaction between modalities, as depicted in Fig.1. For instance, LAVISH [29] uses shared encoders to explore task-specific multi-modal information with a specially designed adapter. DG-SCT [5] introduces a dual-guidance attention mechanism for extracting features across spatial, channel, and temporal dimensions. PFM [26] proposed an efficient and flexible multi-modal fusion approach, which employ cross-attention operation in pretrained encoder layers to learn mutual interactions. However, these approaches have limitations. First, current models primarily

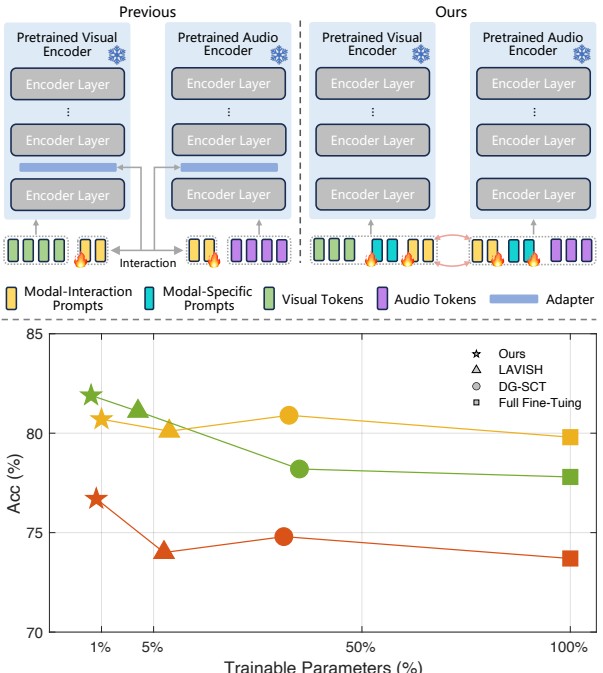

**Figure 1: Comparison of CoPL with previous methods. The top shows that we achieve both modal-specific and modal-interaction fine-tuning, whereas previous methods only focus on multi-modal interaction. The bottom illustrates the efficiency and effectiveness of the proposed CoPL. Different colors represent results on AVE (green), AVS (yellow), and AVQA (red) tasks, respectively.**

focus on fusing multi-modal features, neglecting the crucial aspect of modal-specific fine-tuning for multi-modal tasks. Multi-modal tasks rely on each modality to provide complementary and unique insights in different scenarios. While frozen encoders provide uni-modal information, they are pretrained on single datasets, limiting their ability to provide task-specific cues for various audio-visual tasks. Second, shared prompts are employed for fine-tuning all samples for a given specific task. Due to the limited parameters and expressive capability of prompts, they are unable to adequately represent diverse sample features, resulting in a lack of instance-level feature fine-tuning. Third, they lack the alignment of multi-modal features, a factor proven crucial for multi-modal learning [6, 27].

To address the above limitations, we propose parameter-efficient **Co**llaborative **P**rompt **L**earning **(CoPL)** for transferring pretrained uni-modal models to audio-visual tasks. CoPL achieves remarkable performance with a significantly reduced number of trainable parameters, as illustrated in Fig.1. Specifically, vanilla prompts are divided into Modal-Specific Prompts and Modal-Interaction Prompts. Modal-specific prompts, including video-level prompts and frame-level prompts, explore peculiar cues of each modality in various scenarios to achieve modal-specific fine-tuning. Video-level prompts extract features from the entire video, while local frame-level prompts are appended to each frame to capture local

spatio-temporal associations. Modal-interaction prompts are tailored to transfer information between modalities for multi-modal fusion. They are added only to the last few layers of the pretrained model to further enhance efficiency. Modal-specific prompts and modal-interaction prompts collectively enhance the transfer performance of pretrained uni-modal models on audio-visual downstream tasks. To learn fine-grained instance-level features, we introduce prompt bank-based mutual coupling. The prompt bank consists of randomly initialized prompts. Modal-interaction prompts are adaptively matched with different samples from the prompt bank. The mutual coupling leverages the pretrained model to enhance modal-interaction prompts achieving multi-modal fusion. Additionally, a consistency constraint is introduced to align multi-modal representations, alleviating data distribution bias in uni-modal models. We employ a dual-stream architecture to ensure model flexibility, allowing the backbone to be easily replaced with various pretrained foundation models. The proposed method is evaluated on audio-visual event localization [39], audio-visual segmentation [52], and audio-visual question answering [24] tasks. The main contributions of this work can be summarized as follows:

- Our proposed parameter-efficient Collaborative Prompting Learning (CoPL) decompose vanilla prompts to modal-specific prompts and modal-interaction Prompts for achieving both modal-specific fine-tune and multi-modal fusion.
- Prompt bank-base mutual coupling adequately utilize the generalization ability of pretrained model to achieve multi-modal fusion. This strategy also enhances the model's expression ability to extract fine-grained instance-level features.
- Extensive experiments are conducted across various tasks to demonstrate the effectiveness and efficiency. Our proposed model achieve competitive results with minimum additional trainable parameters compared to previous methods.

## 2  Related Work
## 2.1  Audio Visual Learning
Audio-visual learning aims to explore the association and complementarity between the audio and visual to achieve multi-modal perception. By jointly processing audio and visual inputs, audio-visual learning can enhance the performance and generalization ability of models [1, 32, 40, 42, 48, 51]. Audio-visual learning has applications in a wide range of fields, such as audio-visual event localization, audio-visual segmentation, visual sound localization, and audio-visual question answering [43, 55]. For instance, audio-visual event localization aims to detect events occurring simultaneously in both visual and audio streams within a video [7, 8, 39, 46]. The goal of visual sound localization is to localize the visual regions that emit sound [30, 35–37], while audio-visual segmentation achieves pixel-level localization of sound-emitting objects [10, 13, 52]. Audio-visual question answering is an emerging task that seeks to answer given questions through the integration of visual, audio, and their associations [21, 24, 49]. However, these tasks usually extract features from pretrained models, primarily focusing on designing fusion strategies. The lack of early fusion hinders the effective capture of associations and complementarity between visual and audio. Our proposed method freezes the pretrained models and achieves early fusion through additional prompts.

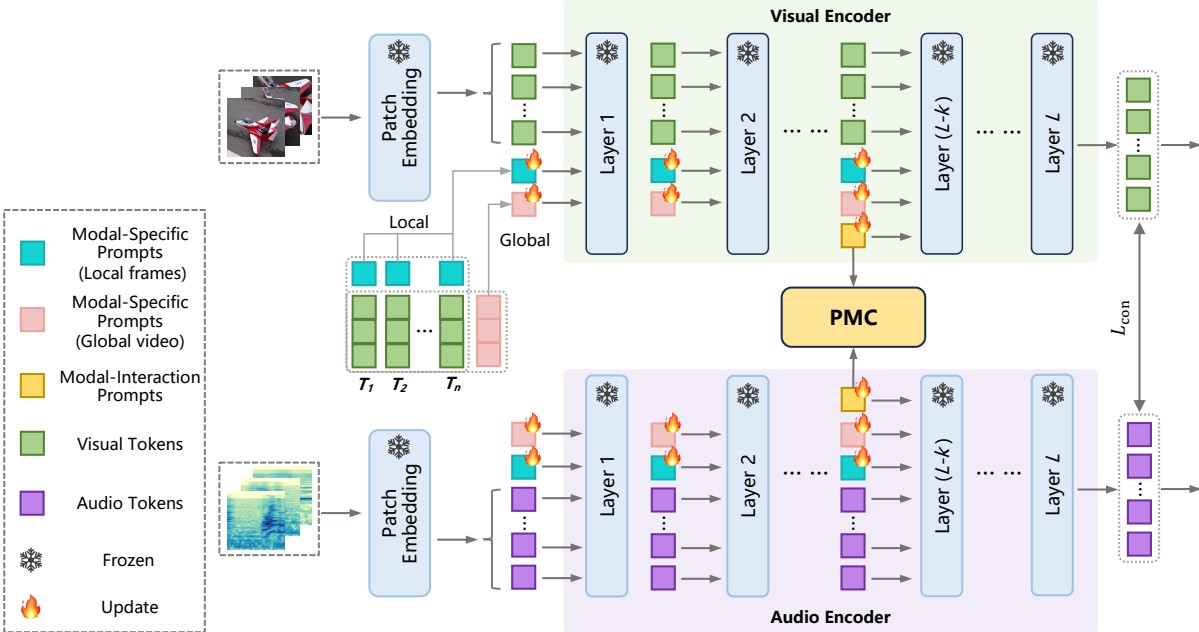

**Figure 2: Overview of the CoPL framework. The audio and visual input are initially projected into token sequences via patch embedding. These input tokens are then concatenated with trainable collaborative prompts, consisting of modal-specific and modal-interaction prompts. The modal-specific prompts include global prompts for video-level feature extraction and local prompts tailored for frame-level features. Modal-interaction prompts are employed to transfer information between modalities through prompt bank-based mutual coupling (PMC).**

## 2.2 Parameter-Efficient Fine-Tuning

Parameter-Efficient Fine-Tuning (PEFT) aims to adapt large-scale pretrained foundation models to downstream tasks for competitive performance with minimal additional trainable parameters [12, 25, 44, 45]. The current strategies for PEFT primarily encompass methods such as Adapter [9, 16, 38], Prompt [18, 22, 53], and LoRA [17]. The adapter approaches introduce additional branches to transfer large pretrained models to different downstream tasks. Prompt customizes the model for specific task by concatenating trainable prompts with input, while LoRA focuses on learning a low-rank factorization of the model's weights. Currently, PEFT in multi-modal learning is predominantly concentrated in the visual-language domain [19, 20, 26, 27], with relatively limited exploration in audio-visual field. The research of PEFT in the audio-visual domain is just beginning to take off. LAVISH first employed a shared encoder for both video and audio by specially designed additional adapter [29]. They employ shared prompts for all samples which limits the model's ability to generalize on downstream tasks. DG-SCT introduced a dual-guidance attention mechanism to fuse audio and visual [5]. They utilizes one modality to guide the feature extraction of counterpart modality across spatial, channel, and temporal dimensions. But the attention mechanisms unavoidably introduce a substantial number of trainable parameters. This render it incapable of alleviating computational costs effectively. Furthermore, these methods primarily focus on fusing multi-modal features, neglecting the crucial aspect of modal-specific fine-tuning for multi-modal tasks. We propose a more efficient prompt-based fine-tuning

method that can generalize the pretrained unimodal model to audio-visual tasks.

## 3 Approach

Our objective is to transfer large-scale pretrained uni-modal models to audio-visual downstream tasks while maintaining a limited number of trainable parameters. Our proposed method, parameter-efficient collaborative prompt learning (CoPL), achieves both modal-specific and modal-interaction fine-tuning for specific task. The overall architecture is depicted in Fig. 2. To enhance model flexibility, we adopt a two-stream structure that integrates pretrained visual and audio encoders within a multi-modal framework. Notably, the audio and visual encoders are interchangeable. Previous studies have demonstrated the effectiveness of ImageNet-Pretrained encoders in extracting audio features [3, 29]. In our experiments, we validate the performance of both shared and unshared encoders. For instance, we utilize SwinTransformerV2 [31] as both the visual and audio encoder, as well as SwinTransformerV2 as the visual encoder in combination with HTS-AT [3] as the audio encoder.

We begin by revisiting the uni-modal transformer-based encoder in Section 3.1, followed by a detailed exposition on how our collaborative prompts transfer uni-modal models to multi-modal tasks in Section 3.2. Subsequently, Section 3.3 introduces the prompt bank-based mutual coupling, which facilitates efficient multi-modal interaction. Finally, a consistency constraint is introduced aimed at guiding the learnable prompts to achieve alignment between audio and visual features. Detailed explanations will follow.

### 3.1 Revisiting Unimodal Transformer

This work primarily considers transformer-based pretrained models as both visual and audio encoders. Given a video input, we initiate the process by extracting visual frames and audio streams. For visual frames $V \in \mathbb{R}^{T \times H \times W \times 3}$, each frame is cut into $m$ non-overlapping patches, which are subsequently projected into visual embeddings denoted as $E_v = \{E_v^1, E_v^2, \cdots, E_v^m\}$, $E_v \in \mathbb{R}^{m \times d_v}$. Regarding audio, we transform the 1-D audio stream into a 2-D spectrogram represented as $\{A_t\}_{t=1}^T$, where $A \in \mathbb{R}^{L \times F}$ with time $L$ and frequency $F$. Subsequently, the audio spectrogram is divided into $n$ patches and transformed into audio embeddings $E_a = \{E_a^1, E_a^2, \cdots, E_a^n\}$, $E_a \in \mathbb{R}^{n \times d_a}$. Assuming the pretrained transformer-based visual and audio encoders are denoted as $T_v$ and $T_a$, respectively. The visual and audio embeddings are fed into the transformer encoder layer to extract features. This process is expressed as follows:

$$
\begin{aligned}
Z_v^{l+1} &= T_v^l(Z_v^l) \\
Z_a^{l+1} &= T_a^l(Z_a^l)
\end{aligned}
\tag{1}
$$

Here, $l \in (1, L)$ denotes the layers of transformer. Each transformer layer $T_v^l$ and $T_a^l$ consists of a stack of multi-head self-attention (MSA) and feed-forward network (FFN) [4, 31]. The features extracted by each transformer layer are represented as $Z_v^l$ and $Z_a^l$, with $Z_v^0 = E_v$ and $Z_a^0 = E_a$ denoting the features after patch embedding.

During the training stage, we keep the pre-trained $T_v$ and $T_a$ frozen and introduce additional learnable prompts to transfer pre-trained model to various downstream tasks. The detailed explanation of introducing collaborative prompts is provided in Sections 3.2 and 3.3.

### 3.2 Efficient Collaborative Prompting

The collaborative prompts consist of modal-specific prompts and modal-interaction prompts, where all prompts are continuous learnable parameters. The modal-specific prompts assist in exploring particular information of each modality for specific audio-visual downstream tasks, while modal-interaction prompts are used to transfer information between modalities for fusing multi-modal information. The modal-specific prompts and modal-interaction prompts enhance the transfer performance through modal-specific fine-tuning and multi-modal fusion. A detailed illustration is presented in Fig.2.

**Modal-Specific Prompts.** Modal-specific prompts, represented as $M^l = \{M_a^l, M_v^l\}$, consist of a set of learnable parameters tailored to audio and visual modalities, respectively. Given specific multi-modal task, modal-specific prompts fine-tune each modality to extract task-specific information. In audio-visual downstream tasks, it's crucial to explore both global semantic information of video clips and local temporal associations among video frames in fine-tuning the pretrained model. To achieve this, modal-specific prompts are designed to include global video-level prompts and local frame-level prompts. For brevity, visual and audio modalities are denoted by subscripts $v$ and $a$, while superscripts $g$ and $f$ represent global video clip and local frame prompts.

Global video-level prompts, defined as $M^g = M_a^g, M_v^g$ for audio and visual, respectively, have dimensions $M_a^g \in \mathbb{R}^{L_g \times d_a}$ and $M_v^g \in \mathbb{R}^{L_g \times d_v}$, where $L_g$ represents the length of global prompts. The

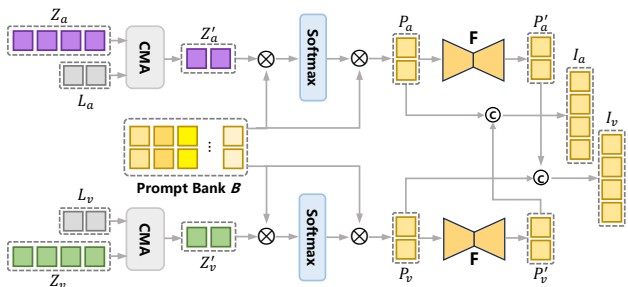

**Figure 3: Illustration of prompt bank-based mutual coupling (PMC).**

global prompts are directly appended with the input video clip to learn features at a holistic level.

Local frame-level prompts are denoted as $\{M_a^f, M_v^f\}$, where $M_a^f \in \mathbb{R}^{L_f \times d_a}$, $M_v^f \in \mathbb{R}^{L_f \times d_v}$, $L_f$ is set to be the same as the length of the video sequence $T$. They are attached with each video sequence $T_i$ to emphasize intricacies within the video.

All these prompts are randomly initialized. Modal-specific prompts are obtained by combining global and local prompts using Eqn.(2), where '[]' represents concatenation.

$$
M_v^l = \left[ M_v^g, M_v^f \right] \qquad M_a^l = \left[ M_a^g, M_a^f \right]
\tag{2}
$$

**Modal-Interaction Prompts.** Our framework employs pre-trained uni-modal models as the encoder, making effective multi-modal fusion crucial for multi-modal tasks. Modal-interaction prompts, denoted as $I^l = \{I_a^l, I_v^l\}$, facilitate the transfer of information between two modalities. $I^l$ is generated from the prompt bank to achieve mutual interaction, as detailed in Section 3.3. Unlike modal-specific prompts $M^l$, which are appended to all encoder layers, modal-interaction prompts $I^l$ are only concatenated to the last $k$ layers of the encoder due to the deeper layers having larger receptive fields and capturing high-level features.

After constructing modal-specific and modal-interaction prompts, we concatenate them with visual and audio embeddings and input them into the transformer layer, as illustrated below:

$$
\begin{aligned}
Z_v^{l+1} &= T_v^l \left[ Z_v^l, M_v^l \right] \\
Z_a^{l+1} &= T_a^l \left[ Z_a^l, M_a^l \right]
\end{aligned}
\qquad if \quad l < k
\tag{3}
$$

$$
\begin{aligned}
Z_v^{l+1} &= T_v^l \left[ Z_v^l, M_v^l, I_v^l \right] \\
Z_a^{l+1} &= T_a^l \left[ Z_a^l, M_a^l, I_a^l \right]
\end{aligned}
\qquad if \quad l \geq k
\tag{4}
$$

The representations $Z_a$ and $Z_v$ from the final encoder layer are utilized for audio-visual downstream tasks.

### 3.3 Prompt Bank-based Mutual Coupling

Audio-visual downstream tasks often involve diverse samples, and existing methods commonly utilize shared prompts for fine-tuning these samples. However, due to the limited capacity of prompts, shared prompts may inadequately represent all samples. In essence, previous approaches focus on task-level fine-tuning, disregarding instance-level feature extraction. To address this limitation, we

introduce prompt bank-based mutual coupling to extract more fine-grained instance-level features, as illustrated in Fig.3.

This process commences by constructing two latent tokens $L_a$ and $L_v$. We employ cross-modal attention (CMA) from [29] to compress the feature $Z_v$ and $Z_a$ in current layer $l$ into $Z'_v$ and $Z'_a$. This process is defined as:

$$Z'_a = L_a + Softmax(L_a Z_a^\top) Z_a$$
$$Z'_v = L_v + Softmax(L_v Z_v^\top) Z_v \qquad (5)$$

Here $L_a \in \mathbb{R}^{b \times d_a}$ and $L_v \in \mathbb{R}^{b \times d_v}$ are randomly initialized.

The prompt bank, denoted as $B = \{P_{b1}, P_{b2}, \ldots, P_{bq}\}$, is introduced, which consists of $q$ randomly initialized prompts. Next, we choose prompts from the bank that are semantically similar to $Z'_v$ and $Z'_a$. Specifically, we calculate the similarity between $Z'$ and each prompt $P_{bi}$ in prompt bank, where the similarity is defined as:

$$e_a^i = \frac{exp(Z'_a P_{bi})}{\sum_{i=1}^{q} exp(Z'_a P_{bi})} \qquad e_v^i = \frac{exp(Z'_v P_{bi})}{\sum_{i=1}^{q} exp(Z'_v P_{bi})} \qquad (6)$$

The similarity score serves as the coefficient for performing a weighted average of the prompts in the bank, yielding $P_v \in \mathbb{R}^{L_I \times d_v}$ and $P_a \in \mathbb{R}^{L_I \times d_a}$.

$$P_a = \sum_{i=1}^{q} e'_a P_{bi} \qquad P_v = \sum_{i=1}^{q} e'_v P_{bi} \qquad (7)$$

Subsequently, $P_v$ and $P_a$ undergo transformation via a mapping layer $F$ to achieve mutual coupling, yielding counterpart prompt representations $P'_v$ and $P'_a$. This transformation is represented as:

$$P'_v = F(P_v) \qquad P'_a = F(P_a) \qquad (8)$$

Here, $P'_v \in \mathbb{R}^{L_I \times d_a}$ and $P'_a \in \mathbb{R}^{L_I \times d_a}$. The mapping function $F$ is defined as $F = \text{Linear}(\text{ReLU}(\text{Linear}(\cdot)))$. It is worth noting that we utilize an efficient bottleneck network as the mapping layer $F$, which notably reduces the number of trainable parameters while effectively transferring information between modalities.

Lastly, we concatenate $P_v$ with $P'_a$ to generate $I_v$, and $P_a$ with $P'_v$ to generate $I_a$, expressed as:

$$I_v = [P_v, P'_a] \qquad I_a = [P_a, P'_v] \qquad (9)$$

Note that the length of $I_v$ and $I_a$ are $2L_I$. Previous methods rely on attention mechanisms to integrate audio-visual features, leading to a considerable increase in trainable parameters [5]. Utilizing shared prompts for all samples ignores instance-level feature extraction. Our prompt bank-based mutual coupling design optimally utilizes the pretrained multi-head attention mechanism to fuse multi-modal information. Furthermore, by extending the vanilla prompts to the prompt bank, we achieve simultaneous task-level and instance-level fine-tuning, significantly enhancing the model's expressive capability. During the training phase, the visual and audio encoders $T_v$ and $T_a$ are frozen, with only the extra prompts and the mapping layer $F$ undergoing training. This approach ensures the efficiency of the model.

### 3.4 Consistency Constraint for Prompting

Despite the utilize of prompts to fine-tune pretrained models for multi-modal tasks, the persistent challenge of feature misalignment between audio and visual persists. To address this issue, we introduce a simple yet effective consistency constraint aimed at enhancing the consistency of representations across different modalities in the feature space. Specifically, we employ the Mean Squared Error (MSE) loss to align the embeddings $Z_a^L$ and $Z_v^L$ from the final layer of audio and visual encoders.

$$L_{con} = \|Z_v^L, Z_a^L\|_2^2 \qquad (10)$$

The consistency constraint facilitates the feature alignment between visual and audio, while enabling prompts to adapt pretrained models to new downstream tasks.

In various audio-visual downstream tasks, the employed training loss functions are diverse. Assuming the downstream task $T$, we define the training loss function as $L_T$. The constraint term, $L_{con}$, is appended with the $L_T$ with a hyperparameter $\lambda$. The final loss $L_F$ is defined as:

$$L_F = L_T + \lambda L_{con} \qquad (11)$$

## 4 Experiments

### 4.1 Downstream Tasks

**Audio Visual Event Localization (AVE)** task aims to detect the audio-visual event that is both visible and audible throughout multiple segments in a video. We evaluate the model on AVE dataset, which contains 4143 videos, and each video last 10 seconds [39]. Following [29], we extract audio and visual features with prompt-base backbone, then we concatenate the audio and visual features and attach a linear layer to obtain the final audio-visual event prediction. The fraction of correctly predicted segments is regarded as the evaluation metric.

**Audio-Visual Segmentation (AVS)** aims to output a pixel-level map of the objects that produce sound in the image frame. The AVSBench-S4 dataset [52] is used for evaluation, which contains 4932 videos with manually annotated pixel-wise annotations of audible objects. We combine our prompt-based feature extractor with the original AVS model. The mean Intersection-over-Union (mIoU) of the predicted segmentation and the ground truth masks is used as evaluation protocol.

**Audio-Visual Question Answering (AVQA)** is an emerging task that aims to provide answers by integrating visual content, audio streams, and their associations within given videos. The MUSIC-AVQA dataset [24] comprises 9288 videos across 22 different musical instruments while covering various question types, including visual, audio, and audio-visual questions. We employ the pretrained text encoder for extracting features of questions. The accuracy calculated by comparing the predictions with the ground truth is used for evaluation.

### 4.2 Implementation Details

For a fair comparison, we process the input video in a similar way with LAVISH and DG-SCT. For all downstream tasks, we extract one frame per second, and the accompanying audio is converted into a 2D spectrogram. The $\lambda$ in Eqn.(11) is set to 0.005 for all experiments, and the length of the prompt bank $q$ is set to 10. Various values of $\lambda$ and $q$ are compared in ablation study. Modal-interaction prompts are expanded to the last 4 encoder layers for ViT and appended to the last 2 layers for SwinTransformerV2. We employ the Adam

**Table 1: Results on Audio-Visual Events Localization. We only consider the additional trainable parameters added to the backbone for comparison purposes. Employing the same downstream network for all methods ensures consistency in parameters in the downstream part. †indicates reproduction using same downstream network for a fair comparison. Our approach achieves optimal results, whether employing shared or unshared encoders.**

| Method | Visual Encoder | Audio Encoder | Additional Params (M) | Total Params (M) | Acc |
|--------|----------------|---------------|----------------------|------------------|-----|
| AVEL | ResNet-152 | VGGish | N/A | 136.0 | 74.0 |
| AVSDN | ResNet-152 | VGGish | N/A | 140.3 | 75.4 |
| CMRAN | ResNet-152 | VGGish | N/A | 148.2 | 78.3 |
| CMBS | ResNet-152 | VGGish | N/A | 216.7 | 79.7 |
| LAVISH | Swin-V2-L | HTS-AT | 6.5 | 270.2 | 78.6 |
| DG-SCT† | Swin-V2-L | HTS-AT | 187.1 | 448.4 | 78.3 |
| Ours | Swin-V2-L | HTS-AT | **0.7** | 263.3 | **79.6** |
| LAVISH | ViT-L-16 (shared) | | 13.5 | 340.7 | 78.1 |
| LAVISH | Swin-V2-L (shared) | | 8.5 | 238.8 | 81.1 |
| Ours | ViT-L-16 (shared) | | 5.0 | 332.8 | 79.2 |
| Ours | Swin-V2-L (shared) | | **1.7** | 232.3 | **81.9** |

**Table 2: Results on Audio-Visual Segmentation. Comparisons on AVSBench-S4 dataset with the mean intersection over union (mIoU) as metric. Our method achieves comparable results with minimal trainable parameters.**

| Method | Visual Encoder | Audio Encoder | Additional Params (M) | Total Params (M) | mIoU |
|--------|----------------|---------------|----------------------|------------------|------|
| AVS | PVT-V2 | VGGish | 82.3 | 174.5 | 78.7 |
| LAVISH | Swin-V2-L | HTS-AT | 17.1 | 297.1 | 78.0 |
| DG-SCT | Swin-V2-L | HTS-AT | 196.6 | 521.4 | **80.9** |
| Ours | Swin-V2-L | HTS-AT | **1.8** | 283.2 | 80.1 |
| LAVISH | ViT-L-16 (shared) | | 27.1 | 375.5 | 74.1 |
| LAVISH | Swin-V2-L (shared) | | 18.3 | 266.4 | 80.1 |
| Ours | ViT-L-16 (shared) | | 4.7 | 353.4 | 76.6 |
| Ours | Swin-V2-L (shared) | | **2.8** | 253.0 | **80.7** |

optimizer in experiments, setting the learning rate for additional prompt-based trainable parameters in the backbone to 1e-3, while setting trainable parameters in downstream network to 1e-5, 5e-5, and 1e-4 for the AVE, AVS, and AVQA tasks, respectively. All experiments are conducted on NVIDIA 4090Ti GPUs.

## 4.3 Results and Analysis

**Audio-Visual Event Localization.** The compared methods include models that solely train downstream networks (e.g., AVEL [39], AVSDN [28], CMRAN [50] CMBS [46]) and fine-tuning paradigms based on pretrained transformers (e.g., LAVISH [29] and DG-SCT [5]). For a fair comparison, we maintain downstream network as a classification layer in all fine-tuning paradigms. Results in Table 1 show that our proposed method achieves optimal performance, whether using shared or unshared encoders. With unshared encoders, CoLP achieves a 1.3 improvement compared to DG-SCT, while with shared encoders, our model outperforms LAVISH by 0.8. Employing a shared encoder performs better compared to unshared encoders, primarily due to the larger-scale Swin-Transformer extracting more robust features than HTS-AT. Notably, LAVISH requires 8.5M additional parameters, while DG-SCT has 187M. In

contrast, our method requires only **1.7M** additional parameters with shared encoder, while **0.7M** additional parameters with unshared encoder, effectively reducing the demand for computational resources.

**Audio-Visual Segmentation.** The evaluation results for the Audio-Visual Segmentation (AVS) task are presented in Table 2. Compared to the AVS approach [52], which primarily focuses on training downstream networks, our method achieves a notable improvement in performance by 2.0 while significantly reducing the number of additional parameters required (2.8M vs. 82.3M). In contrast to fine-tuning paradigms, our approach achieves competitive performance levels comparable to those of LAVISH [29] and DG-SCT [5], despite requiring substantially fewer additional parameters. Specifically, LAVISH necessitates an additional 18.3M parameters, and DG-SCT requires 196.6M additional parameters. In contrast, our method achieves comparable performance with only **2.8M** additional parameters when using a shared encoder and **1.8M** additional parameters for unshared encoders. These empirical findings affirm comprehensively that our model achieves competitive results compared to state-of-the-art models, leveraging fewer training parameters. Notably, there is a margin when utilizing ViT

**Table 3: Results on Audio-Visual Question Answering. All models are evaluated on on MUSIC-AVQA dataset. We report the accuracy of audio question (AQ), visual question (VQ), and audio-visual question (AVQ) respectively.**

| Method | Visual Encoder | Audio Encoder | Additional Params (M) | Total Params (M) | AQ | VQ | AVQ | Avg |
|---|---|---|---|---|---|---|---|---|
| AVSD | VGG-19 | VGGish | N/A | N/A | 68.5 | 70.8 | 65.5 | 67.4 |
| AVST | ResNet-18 | VGGish | 18.5 | 94.4 | 74.1 | 74.0 | 69.5 | 71.5 |
| PSTP | CLIP- ViT-B/32 | VGGish | 4.3 | N/A | 70.9 | 77.3 | 72.6 | 73.5 |
| LAVISH | Swin-V2-L | HTS-AT | 12.9 | 290.5 | 75.4 | 79.6 | 70.1 | 73.6 |
| DG-SCT | Swin-V2-L | HTS-AT | 186.3 | 513.3 | 77.4 | 81.9 | 70.7 | 74.8 |
| Ours | Swin-V2-L | HTS-AT | **0.7** | 271.1 | 76.8 | 77.1 | 75.2 | **75.9** |
| LAVISH | ViT-L-16 (shared) | | 27.1 | 362.5 | 74.1 | 73.6 | 74.7 | 74.4 |
| LAVISH | Swin-V2-L (shared) | | 17.1 | 255.3 | 75.7 | 80.4 | 70.4 | 74.0 |
| Ours | ViT-L-16 (shared) | | 5.0 | 340.7 | 75.1 | 77.2 | 75.0 | 75.6 |
| Ours | Swin-V2-L (shared) | | **1.8** | 240.2 | 77.3 | 77.6 | 76.3 | **76.7** |

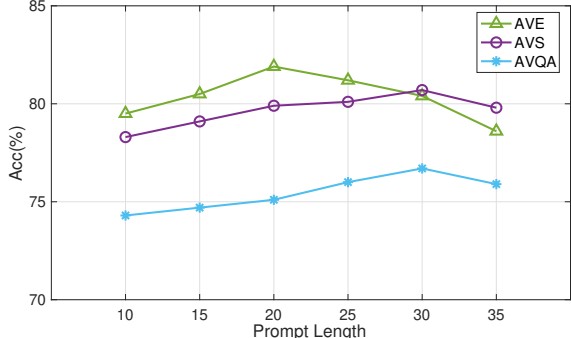

**Figure 4: Impact of prompts length**

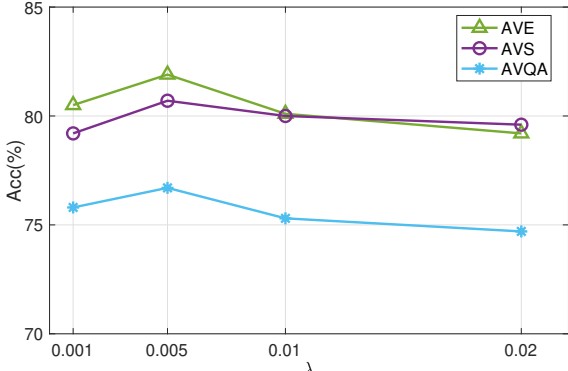

**Figure 5: Impact of $\lambda$**

as a shared encoder compared to Swin-V2. This disparity is primarily attributed to the standard ViT model's lack of multi-scale information capabilities, which are crucial for the AVS task.

**Audio-Visual Question Answering.** On the MUSIC-AVQA dataset, we conducted a comparative analysis, the results are presented in Table 3. Compared to PSTP [23] which only train downstream network, our model exhibits a significant performance improvement (76.7 compared to 73.5). Although they employ powerful CLIP to extract visual features, our approach achieves middle fusion in feature extraction which result in better performance. In comparison to fine-tuning paradigms, our model still achieves the best results whether using shared or unshared encoder, accompanied by a substantial reduction in the number of parameters used (1.8M vs. 17.1M vs. 186.3M). On the most challenging audio-visual questions type, the proposed CoPL achieves significant improvement, which proves that our model effectively fuse multi-modal.

The comprehensive experimental results on audio-visual downstream tasks indicate that our model achieves both modal-specific and modal-interaction tuning, efficiently transferring from pretrained unimodal models to audio-visual downstream tasks.

## 4.4 Ablation Studies

*4.4.1 Analysis of Shared and Unshared Encoder.* The flexibility of our framework facilitates the straightforward replacement of the

backbone with different encoders. We assess different pretrained models as audio encoders, including HTS-AT pretrained on AudioSet and SwinTransformerV2 pretrained on ImageNet, with results presented in Table 1-3. Despite HTS-AT pretrained on AudioSet being capable of extracting exclusive audio features, the results indicate that using shared SwinTransformerV2 for both audio and visual encoders yields superior performance. This is attributed to the significantly higher capacity of SwinTransformerV2, which incorporates 24 Transformer blocks, while HTS-AT only includes 12 blocks. It is noteworthy that the shallow backbone (12 transformer blocks in HTS-AT) further reduces the trainable parameters (0.7M vs. 1.7M on AVE, 1.8M vs. 2.8M on AVS, and 0.7M vs. 1.8M on AVQA), with only a slight degradation in performance.

*4.4.2 Analyzing Each Proposed Module.* In this section, we conduct ablation experiments to investigate the effect of each proposed module on the AVE dataset. We utilize SwinTransformerV2 as the shared encoder for both visual and audio modalities in all ablation experiments. The results, depicted in Table 4, utilize $M$ to denote modal-specific prompts, which encompass video clip level $M^g$ and frame level $M^f$, while *PMC* denotes the prompt bank-based mutual coupling, which includes modal-interaction prompts $I$ and prompt bank. The $L_{con}$ corresponds to the consistency constraint

**Table 4: Ablation Study. We assess the effectiveness of each module in the proposed model on AVE task.** $M$ **indicates modal-specific prompts, while** $PMC$ **represents the prompt bank-based mutual coupling module, which consists of modal-interaction prompts** $I$ **and prompt bank.**

| $M$ | | $PMC$ | | $L_{con}$ | Acc |
|---|---|---|---|---|---|
| $M^g$ | $M^f$ | $I$ | Prompt Bank | | |
| | | | | | 77.6 |
| ✓ | | | | | 78.7 |
| ✓ | ✓ | | | | 79.2 |
| ✓ | ✓ | ✓ | | | 80.5 |
| ✓ | ✓ | ✓ | ✓ | | 81.4 |
| ✓ | ✓ | ✓ | ✓ | ✓ | 81.9 |

in Eqn.(10). The results substantiate that all proposed modules significantly enhance the model's performance. Specifically, incorporating modal-specific prompts and modal-interaction prompts separately into the backbone increases the accuracy by 1.6 and 1.3, respectively. For modal-specific prompts, composed of $M^f$ and $M^g$ compared to only $M^g$, yields an additional 0.5 improvement. Introducing a prompt bank to extract instance-level features results in an improvement of 0.9. Lastly, the appended consistency constraint leads to a further 0.6 increase in the model's performance. It is noteworthy that all introduced modules contribute less than 1% additional parameters, considerably fewer than in previous models, thus demonstrating the efficiency and effectiveness of our proposed method.

*4.4.3 Impact of Prompt Length.* This section investigates how the length of prompt tokens affects the performance. We have devised two distinct types of prompts: modal-specific prompts $M$ and modal-interaction prompts $I$. The local prompts $M^f$ of modal-specific prompts aim to explore frame-level details, so we align its length with the input video sequence. For AVE and AVQA tasks, this length is set to 10, while for AVS tasks, it is set to 5. Experiments are conducted with lengths of 10, 15, 20, 25, 30, and 35 for both $M^g$ and $I$, with the same length set for these two prompts. The results, illustrated in Fig.4, demonstrate that setting the lengths of $M^g$ and $I$ to 20 for AVE and 30 for AVS and AVQA achieves optimal model performance. It is noteworthy that, performance tends to degrade when the length is too short. This limitation arises from the prompts having an extremely limited number of trainable parameters, making it easy for the model to approach the generalization limit. Conversely, too many prompts may harm the performance due to over-fitting. Therefore, selecting the appropriate prompt length for various downstream tasks is essential.

*4.4.4 Effect of Consistency Constraint.* We introduce consistency constraints for aligning audio and visual features. The loss in Eqn.10 is employed as an additional constraint for multi-modal downstream tasks. In this section, we investigate the effect of consistency constraint loss on the model. Specifically, $\lambda$ in Eqn.11 is employed to append the consistency constraint to task-specific loss. We conduct ablation study on coefficient $\lambda$, the results are shown in Fig.5 It can be seen that proper consistency constraint loss is crucial. Large

**Table 5: Comparison of trainable parameters and training memory usage on AVE task.**

| Method | Trainable Params (M) | Training Memory (GB) | Acc |
|---|---|---|---|
| LAVISH | 8.5 | 18.9 | 81.1 |
| DG-SCT | 187.1 | 19.9 | 78.3 |
| Ours | 1.7 | 15.4 | 81.9 |

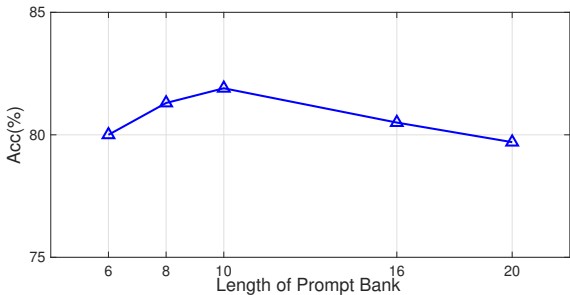

**Figure 6: Impact of prompt bank length**

constraints would lead the model to pay attention to alignment during training, while small constraint has limited effect. The model achieves optimal performance when $\lambda$ is set to 0.005.

*4.4.5 Analysis of Prompt Bank.* In this section, we evaluate the length $q$ of the prompt bank. The experiments are conducted on the AVE task, and the length $q$ varies from 6 to 20. The results in Fig.6 show that setting the $q$ to 10 achieves optimal performance. The model performance improves with an increase in $q$, but when $q$ is too large, performance will decline due to reaching saturation.

*4.4.6 Efficiency Analysis.* We analyze the additional trainable parameters and training memory to evaluate the efficiency of the model. Both LAVISH and our model utilize the Shared SwinTransformerV2 as the backbone, while DG-SCT utilizes SwinTransformerV2 for visual encoding and HT-SAT for audio encoding. Experiments are conducted on the AVE task, and the results are presented in Table 5. Our model reduces the trainable parameters by more than 4× times compared to LAVISH, with minimal training memory requirements. This demonstrates the efficiency of the proposed model.

## 5 Conclusion

This paper introduce parameter-efficient Collaborative Prompt Learning (CoPL) that transfer large-scale pretrained uni-modal models to audio-visual downstream tasks. Our model divided vanilla prompts to modal-specific and modal-interaction prompts. The modal-specific prompts enable to tuning each modality for specific tasks, while modal-interaction prompts efficiently transfer information between modalities for multi-modal fusion. The prompt bank-based mutual coupling assists in extracting fine-grained instance-level features. Extensive experiments across various tasks demonstrate the effectiveness and efficiency of our method.

## Acknowledgments

This work was supported by the National Key R&D Program of China under grant 2022YFF0901800, the National Natural Science Foundation of China (NSFC) under grants No. 62072367, 62176205, 62302383, and 62372365, and the Open Project of the Communication University of China under grant JGKFKT2305.

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
