# OpenReview forum: "CoPL:Parameter-Efficient Collaborative Prompt Learning for Audio-Visual Tasks"
_acmmm.org/ACMMM/2024/Conference — MM2024 Poster_

### Official Review · Reviewer_gdSb · 2024-05-14

**Rating:** 4
**Confidence:** 3

**Summary:**

This paper proposes a parameter-efficient tuning method for pre-trained uni-modals for audio-visual tasks. It proposes to fine-tune not only the modal-interaction prompts but also the modal-specific prompts. The modal-interaction prompts aim to effectively fuse the multi-modal information together, while the modal-specific prompts are learned to focus on individual modal tasks. A prompt bank-based method is proposed to enhance the model’s generalization. The experimental results show that their approach can achieve high accuracy with few trainable parameters.

**Strengths:**

1. Clarity and Coherence: The paper exhibits very clear and logically structured writing, effectively presenting the objectives, methodology, and backs their methods with extensive experiments.
2. Innovative Approach: The proposed modal-specific and modal-interaction prompts are important. They not only need to interact with each other but also need to preserve themselves for uni-modal capability. Furthermore, the prompt bank is crucial to mimic instance-level features, which helps the model to generalize well.
3. Comprehensive Evaluation: The authors conduct thorough comparisons with multiple baselines, accompanied by sufficient ablation studies to showcase the effectiveness of each of their proposals, thereby substantiating the effectiveness of their proposed approach.

**Limitations:**

1. Using a prompt bank is a good idea. But other than the ablation study, what information is actually being learned for the prompt inside? I would encourage the authors to visualize the learned prompts inside the bank and correlate them to the instance-level features. This will strengthen their assumption of the prompt bank proposal.
2. The experiments are presented solely with quantitative results. I would encourage the authors to provide qualitative results with some visual examples to help readers understand their strengths.
3. Failure case analysis is missing. The authors are encouraged to discuss and present what does not work for their proposals.

Questions:
1. How do you modify positional encoding for local/global prompts? It seems that for local prompts, the positional information should be preserved as it is assigned to each frame with a specific time instance.
2. Do we really need the mapping function $F$ in the PMC module? Can we fuse/concatenate $P_a$ and $P_v$ directly? I would encourage an ablation study on this.

**Suitability:**

3

---

### Official Review · Reviewer_qCqd · 2024-05-25

**Rating:** 3
**Confidence:** 2

**Summary:**

This paper proposes parameter-efficient Collaborative Prompt Learning(CoPL) to fine-tune both uni-modal and multi-modal features. Experimental results demonstrate its effectiveness on various audio-visual downstream tasks.

**Strengths:**

1. The paper is well-written and easy to follow.
2. The experimental part is very comprehensive. The experiment results are impressive.

**Limitations:**

1. The method of this paper is similar to many methods, including but not limited to [1][2][3]. Although there is some discussion with some methods in the paper (Lines 264-267), merely claiming the difference in application scenarios is not satisfactory.
2. Experimental heuristics are required to set different downstream prompt lengths, which hinders efficient downstream adaptation.

[1] Consistency-guided Prompt Learning for Vision-Language Models.

[2] Maple: Multi-modal prompt learning.

[3] Self-regulating Prompts: Foundational Model Adaptation without Forgetting

**Suitability:**

2

---

### Official Review · Reviewer_Jnjy · 2024-05-25

**Rating:** 5
**Confidence:** 3

**Summary:**

The text discusses the effectiveness of Parameter-Efficient Fine Tuning (PEFT) in transferring foundation models to downstream tasks. It points out that current methods focus on multi-modal fusion, neglecting modal-specific fine-tuning, which is important for multi-modal tasks. The authors propose Collaborative Prompt Learning (CoPL) for fine-tuning both uni-modal and multi-modal features. CoPL uses modal-specific prompts for fine-tuning each modality on specific tasks and modal-interaction prompts for exploring inter-modality association. It also introduces a prompt bank-based mutual coupling to extract instance-level features, enhancing the model's generalization ability. Experimental results show that this approach performs comparably or better on various audio-visual downstream tasks, using approximately 1% extra trainable parameters.

**Strengths:**

1. CoPL decompose vanilla prompts to modal-specific prompts and modal-interaction Prompts for achieving both modal-specific fine-tune and multi-modal fusion.
2. The strategy of prompt bank-based mutual coupling effectively leverages the generalization capability of a pretrained model to accomplish multi-modal fusion. CoPL amplifies the model's expressive power, enabling it to extract detailed instance-level features.
3. CoPL achieve competitive results with minimum additional trainable parameters compared to previous methods.

**Limitations:**

1. Can the PMC module be used in every layer interaction? If so, is it possible to add the appropriate ablation experiments? If not, is it possible to give more detailed explanatory notes?
2. Has there been any consideration of the additional computational effort of adding only the token dimension and a separate PMC module where interactions between modalities may be relatively small?

I would like to recommend receiving this article if my question can be effectively answered.

**Suitability:**

3

---

### Official Review · Reviewer_aPAs · 2024-05-26

**Rating:** 4
**Confidence:** 3

**Summary:**

This paper investigates parameter-efficient fine tuning method for audio-visual tasks. It suggests that the modal-specific fine-tuning is as consequential as multi-modal fusion in the tasks. To this end , this paper proposes collaborative prompt learning (CoPL) to finetune features in both uni-modal and multi-modal ways. Modal-specific prompts are tailored for fine-tuning each modality on specific tasks while the modal-interaction prompts are customized to efficiently explore inter-modality association. Extensive experiments demonstrate that CoPL can effectively reduce training parameters and improving model performance simultaneously.

**Strengths:**

1. The motivation of this paper is well-founded and clear.
2. The experiments are sufficient and have demonstrated the effectiveness of the proposed method.
3. The paper is well-written and presents its concepts clearly and comprehensibly which makes it easy to follow.

**Limitations:**

1. The main innovation component is the Prompt Bank-based Mutual Coupling module(PMC), but there lacks the ablation studies for PMC. Such as the effect of  and .
2. Lack of comparison of model computational complexity(FLOPs) and inference time.
3. The icons of modal-specific prompts in different modal branches in fig.2 should be different.

**Suitability:**

3

---

### Meta-Review · Area_Chair_RTBT · 2024-07-01

**Recommendation:** Accept (Poster)
**Confidence:** 5

**Metareview:**

This paper proposes a parameter-efficient collaborative prompt learning approach for fine tuning both uni-modal and multi-modal models. Initially the paper received 1 weak accept, 2 borderline accepts, and 1 borderline reject. Authors addressed most concerns from reviewers adequately in the rebuttal, while Reviewer gdSb still have concern on the analysis of failure cases. Two reviewers raised their scores after rebuttal. The AC also agrees the paper proposes an interesting idea, and the experimental results are convincing. Therefore, the AC suggests to accept this paper, but authors should place further analysis of failure cases as suggested by Reviewers gdSb.